# Effects of Relative Leisure Deprivation between Regions on Life Satisfaction in Koreans: Focusing on Baby Boomers

**DOI:** 10.3390/ijerph182412905

**Published:** 2021-12-07

**Authors:** Hyang-Hee Hwang, Yu-Jin Lee, Bo-Ram Kim

**Affiliations:** 1Department of Sport Science, Kangwon National University, Chuncheon 24341, Korea; phyhee@kangwon.ac.kr; 2Department of Sport Science, University of Seoul, Seoul 02504, Korea

**Keywords:** baby boomers, relative leisure deprivation (RLD), life satisfaction, place of residence, moderating effect

## Abstract

Regional disparities in leisure facilities deprive residents of opportunities to participate in leisure. This study aimed to provide basic data for establishing public leisure welfare policies to reduce the leisure gap among different regions and to verify the effects of relative leisure deprivation (RLD) on the life satisfaction of Koreans, with a focus on the baby boomer generation. For this purpose, 7 items of demographic characteristics related to gender, age, marital status, job status, residence area, monthly income, and educational background, 18 items of relative leisure deprivation consisting of egoistical, resourceful, cognitive, and emotional leisure deprivation, and 5 items of life satisfaction were investigated. The questionnaire consisted of a total of 30 questions and a mobile survey was conducted in October 2020, and a total of 412 copies were used for the final analysis. The results showed that there were differences in RLD and life satisfaction depending on where the participants lived; RLD (M = 3.21, M = 2.95) was higher and life satisfaction (M = 3.36, M = 3.72) was lower in rural areas, as compared with urban areas. Second, baby boomers’ RLD had a negative effect (β = −0.5391, *p* < 0.001) on life satisfaction. Third, the place of residence moderated (β = 0.5240, *p* < 0.001) the relationship between RLD and life satisfaction; a higher RLD led to a lower level of life satisfaction for baby boomers living in rural areas (95% CI: −0.7369~−0.3413), whereas the RLD of those in urban areas did not affect their life satisfaction. Therefore, central or local governments must effectively narrow the regional gap through a balanced distribution of leisure resources to remote and underdeveloped environments, thereby minimizing the RLD of citizens and seeking improvement in life satisfaction. Finally, the part that the psychological aspect of the individual study was not considered due to the limitations of quantitative research suggests the direction of subsequent research.

## 1. Introduction

Understanding the issue of leisure inequality among different regions is important in reducing regional gaps that deprive residents of many opportunities to participate in leisure activities and improve their quality of life. Townsend [1] presented deprivation as a multidimensional concept that encompasses a lack of adequate leisure in life, as well as a lack of services and physical standards in material, social, and cultural elements of life beyond poverty; thus, deprivation was defined with an emphasis on the relative state of inequality based on the level of activity or experience [2]. Crosby [3,4] argued that individuals have negative emotions when something they feel they deserve is not fulfilled in comparison with others, and they experience relative deprivation when they perceive a discrepancy between what they want and what they actually have. As such, deprivation is a relative and negative emotional state that is perceived based on physical standards and subjective experiences.

Hwang and Kim [5] conceptualized relative leisure deprivation (RLD) as the “subjective emotional state in which individuals feel relative inequality in overall leisure resources” in the relationship between leisure inequality and relative deprivation. The key point of RLD is the emphasis on the relative state instead of an absolute comparison, which is based on subjective emotion toward overall leisure resources. Overall leisure resources refer to universal social services provided by the central or local governments to meet the basic needs and rights of citizens for participation in leisure activities. Social services provided by the central or local governments have a profound effect on the quality of life of residents [6,7,8]. In Korea, the Framework Act on Social Security guarantees that all citizens lead a happy life worthy of human dignity.

Recently, the United Nations Conference on Trade and Development in 2021 raised Korea’s status from a developing economy to a developed one [9]. However, Korea remains in the lower ranks in terms of the Better Life Index (BLI) announced by the Organization for Economic Cooperation and Development (OECD) in May 2021. The BLI is an index of the extent to which individuals enjoy a good life. As another measure for a happy life, the World Health Organization defines quality of life as “an individual’s perception of their position in life in the context of the culture and value systems in which they live and in relation to their goals, expectations, standards, and concerns” [10]. Borthwick-Duffy [11] defined quality of life in three aspects based on individuals’ life conditions: quality, satisfaction, and life satisfaction. Life satisfaction can be considered an overall assessment of quality of life based on the standard chosen by oneself [12].

According to Article 24 of the Universal Declaration of Human Rights, “Everyone has the right to rest and leisure, including a reasonable limitation of working hours and periodic paid holidays”. In Korea, the Framework Act on the Promotion of Leisure of Citizens has been enforced since 21 March 2017; Article 1 (Purpose) of the Act specifies that “The purpose of this Act is to lay foundations for free leisure activities by citizens and to help them improve their quality of life through participating in various leisure activities, by providing for basic provisions related to the establishment and implementation of policies for the promotion of leisure”. Moreover, the details of the Framework Act specify that the central and local governments shall develop and disseminate leisure programs, collect and provide leisure information, provide leisure education, expand leisure facilities and venues, train leisure professionals, and guarantee leisure for all sections of the society, without alienating the disadvantaged groups. In other words, the central or local governments must guarantee citizens’ rights to enjoy leisure, prohibit discrimination regarding the enjoyment of leisure, and guarantee the right to receive leisure education. Despite these institutional grounds based on law, whether they are actually implemented in real life is still questionable.

In Korea, 2020 marks the beginning of the transition of the baby boomers (born 1955–1963 cohort) into the elderly generation. The baby boomer generation is a generation that is not welcomed anywhere between parents and children and is a work-oriented generation who moved to the city for college life and first socialization in the 1980s, due to rapid social changes such as industrialization and urbanization [13,14,15]. Although they want to return hometown after retirement, they report that they feel the relative regional disparities between their current residence and their hometown where they wish to move, reducing their desire to return hometown [16]. Compared to cities, rural areas are not likely to improve public service conditions unless the amount of policy distribution is significantly increased due to the characteristics of the settlement conditions where the population is dispersed over a large area [17]. The environmental characteristics, services, and social and cultural characteristics of the community shared with the local residents affect the health of the residents [18,19].

Many studies have been conducted on regional disparities and deprivation in Korea, most of which are related to health inequality, levels of health, and income inequality [20,21,22,23]. As national development in Korea has been focused on urban areas, it is also necessary to examine the relationship between RLD and quality of life in the rural population, because residents of rural areas may feel alienated from the process of development. Rural areas face disadvantages across all domains of life, such as health, safety, material resources, residential environment, and participation in leisure activities. This is because rural areas in Korea are usually agricultural or mountainous settlements, or fishing villages, where the population is scattered across wide areas [17]. Without active development and increased public services for these areas, they are bound to face disadvantages compared to urban areas with high accessibility.

Korea was ranked 28th out of 35 countries in the BLI (http://www.oecdbetterlifeindex.org/, accessed on 12 July 2021), as disclosed in “How’s Life?” published by the OECD (8 March 2020). Baby boomers in Korea entered retirement without being well-prepared, as the “generation stuck in the middle”, sacrificing themselves between their parents and children. Yoon [24] developed the Leisure Happiness Index to measure satisfaction with having a happy life through participation in leisure activities. The results at the regional level showed that larger regions scored higher on the Leisure Happiness Index. This indicates that residents of big cities experience higher levels of happiness through leisure. The central or local governments are obliged to provide public services for citizens’ desire for leisure, which is supported by the Framework Act on the Promotion of Leisure of Citizens. Baby boomers who become part of the older adult population spend more time in leisure activities and have higher expectations and demands for their leisure time. Some studies show that despite the desire to move to rural areas after retirement, baby boomers are reluctant to do this because of relative regional disparities, which suggests the need to examine the relationship between RLD and quality of life among residents of different regions. Therefore, this study analyzed how baby boomers’ RLD affects life satisfaction, and the moderating effect of place of residence on this relationship, thereby highlighting the need to reduce the leisure gap among regions. Based on the foregoing, the following hypotheses were proposed.

**Hypothesis** **1** **(H1).***There is a difference in RLD and life satisfaction depending on baby boomers’ place of residence*.

**Hypothesis** **2** **(H2).***Baby boomers’ RLD affects their life satisfaction*.

**Hypothesis** **3** **(H3).***Baby boomers’ place of residence moderates the relationship between RLD and life satisfaction*.

## 2. Methods

### 2.1. Participants and Sampling

This study was approved by the institutional review board of the Kangwon National University from the design and planning stages (KWNUIRB-2021-04-005-001).

This study set the population of baby boomers nationwide. In the course of conducting the preliminary and main surveys, the content validity of each questionnaire was verified through a meeting of an expert group consisting of 5 professors and PhDs majoring in leisure/sport science. The preliminary survey was conducted on 100 baby boomers living in rural areas, recruited from local health clinics at the level of eup/myeon/gun (Korean regional units). The suitability of the measurement items was verified by considering the adequacy, clarity, and time required for the survey. In the main survey, the survey was conducted based on the proportional allocation sampling method such as gender, cursed area, and size of residential area. The main survey was commissioned by the research firm Rende Research (www.rende.co.kr, accessed on 12 July 2021) and conducted through mobile phones for a month (October 2020). A total of 500 copies were distributed, and the results of the socio-demographic characteristics of 412 people used in the actual analysis are shown in Table 1.

### 2.2. Validity and Reliability of the Measurement Tool

Following validation of the preliminary survey, a questionnaire was used as the measurement tool to meet the study purpose. To determine whether the items developed based on previous studies sufficiently measured the intended content and were suitable for domestic circumstances, content validity and item suitability were tested by experts, including two professors of physical education and three doctoral researchers in leisure studies. The questionnaire comprised items on the demographic characteristics of baby boomers, as well as items on RLD and life satisfaction. RLD and life satisfaction, which are the key variables of this study, were rated on a five-point Likert scale from “strongly disagree” (1 point) to “strongly agree” (5 points).

First, RLD was measured using the Relative Leisure Deprivation Scale (RLD-S) developed by Hwang, Lee, and Kim [25], which comprises 18 items on four factors: egoistical deprivation, resource deprivation, cognitive deprivation, and emotional deprivation. Some examples are: “I do not have enough time for leisure compared to others”, “The place where I live in does not have enough leisure facilities compared to other regions”, “I am unsatisfied with the fact that I do not have enough money to meet leisure costs”, and “I am furious that the place I live in is disadvantaged because of a lack of leisure programs compared to other regions”. Second, life satisfaction was measured using a questionnaire from the study by Ko [26], comprising five items on a single factor. Some examples are: “I am very satisfied with my life” and “My life is almost ideal in many aspects” (Table 2).

The relationship between latent variables and measurement items was validated through a confirmatory factor analysis. The absolute fit indices used to assess the fit of the model were chi-square/degrees of freedom (χ^2^/*df*), root mean square error of approximation (RMSEA), goodness-of-fit index (GFI), comparative fit index (CFI), and Tucker–Lewis index (TLI). The fit based on the confirmatory factor analysis was χ^2^ = 422.34, *df* = 124, GFI = 0.923, CFI = 0.912, TLI = 0.904, RMSEA = 0.069. Therefore, the results of the confirmatory factor analysis of RLD comprised 18 items and four factors, as shown in Table 3.

Cronbach’s α was calculated to test the reliability of the measurement tool. Cronbach’s α for RLD and life satisfaction was 0.822 and 0.827, respectively, thereby showing relatively high reliability.

### 2.3. Data Processing Method

SPSS 25.0 and AMOS 25.0 were used to test the moderating effect of place of residence on the relationship between RLD and life satisfaction. Frequency analysis and descriptive statistics were used to determine the mean and standard deviation of the variables, as well as the general characteristics of the participants. Confirmatory factor analysis was conducted to test the dimensionality and validity of the factor structure of the variables, and Cronbach’s α was calculated to test reliability. In addition, an independent *t*-test was conducted to verify the difference in RLD and life satisfaction depending on the place of residence (rural/urban areas). The assumption of equal variance was judged by the results of Levene’s *F* test. Pearson’s correlation analysis was conducted to determine the correlation between the variables, and multi-collinearity was identified. To test the moderating effect of place of residence, Model 1 of Process Macro ver.3.4 (Armonk, New York, NY, USA), was used for the analysis. Five thousand bootstrapping clones were applied. The variable (place of residence) in the nominal scale was changed to a dummy variable before the analysis. The statistical significance level was set at 0.05.

## 3. Results

### 3.1. Differences in Relative Leisure Deprivation and Life Satisfaction by Regions

The results of the *t*-test showing the differences in RLD and life satisfaction depending on the place of residence are shown in Table 4. The analysis revealed a statistically significant difference (*t* = 5.982, *p* < 0.001) in RLD depending on the place of residence. Specifically, residents of rural areas (*M* = 3.21) showed a higher level of RLD than residents of urban areas (*M* = 2.95). Further, there was a statistically significant difference (*t* = −6.106, *p* < 0.001) in life satisfaction depending on the place of residence; rural residents (*M* = 3.36) showed a lower level of life satisfaction than their urban counterparts (*M* = 3.72).

### 3.2. Moderating Effect of Regions on the Relationship between Relative Leisure Deprivation and Life Satisfaction

A correlation analysis was conducted among the variables prior to analyzing the moderating effect of place of residence on the relationship between RLD and life satisfaction of baby boomers. The results ranged from −0.289 to 0.282, thereby showing both a positive and negative correlation within the significance level (*p* < 0.01). Moreover, there was no multi-collinearity because the correlation coefficient was no higher than 0.8.

Model 1 of Process Macro ver.3.4 was used to verify the moderating effect of place of residence on the relationship between RLD and life satisfaction of baby boomers. The results of the analysis, by designating 5000 bootstrap replications and setting a confidence interval of 95%, are shown in Table 5. RLD had a negative effect on life satisfaction (*β* = −0.5391, *p* < 0.001), and place of residence had a positive effect on life satisfaction (*β* = 0.2832, *p* < 0.001). The interaction term had a significant effect on life satisfaction (*β* = 0.5240, *p* < 0.001), thereby confirming the moderating effect of place of residence. Moreover, the R^2^ increase due to the interaction of RLD and place of residence was 0.144 (*p* < 0.001), which was statistically significant, thereby verifying the moderating effect of place of residence on the relationship between RLD and life satisfaction.

The conditional effects of how the independent variables affected the dependent variables depending on specific values of the moderating variable are as shown in Table 6; all were statistically significant. Thus, the moderating effect was visualized as shown in Figure 1 to verify the form.

By classifying place of residence into rural (0) and urban areas (1), it was found that the simple slope between RLD and life satisfaction had a significant conditional effect in rural areas, but not in urban areas. That is, RLD had a statistically significant negative effect on life satisfaction in rural areas, whereas it did not affect life satisfaction in urban areas. Baby boomers with higher RLD in rural areas showed lower life satisfaction, whereas the RLD of baby boomers in urban areas did not affect their life satisfaction.

## 4. Discussion

The purpose of this study was to provide basic data for promoting the establishment of public leisure welfare policies to reduce the leisure gap among different regions, and to verify the effects of RLD on the life satisfaction of Koreans who are baby boomers. The following discussion is presented based on the results of this study.

First, RLD and life satisfaction differed depending on the baby boomers’ place of residence. Specifically, RLD was higher in rural than in urban areas, whereas life satisfaction was lower in rural than that in urban areas. In general, various public kinds of infrastructure and services are developed collectively at the regional level, in addition to individual resources [27]. However, these resources tend to be distributed unequally in different areas, which restricts access to the necessary resources and opportunities for residents of deprived areas [28].

In particular, rural areas lack leisure resources such as infrastructure and facilities to participate in leisure activities compared with urban areas, and public services are not likely to show much improvement unless there are changes in welfare policies to significantly increase services distributed [24]. Therefore, the fact that rural areas have a higher sense of RLD than urban areas is somewhat predictable. Oh [29] found that rural areas showed lower income than urban areas, which is a major factor inhibiting participation in physical activities during leisure time. Income inequality is one of the main causes of leisure inequality [30]. Jeong [31] discovered that accessibility to leisure activity participation had a critical effect on relative deprivation and awareness of regional gaps in leisure and culture and emphasized the importance of increased leisure facilities and equal access to leisure opportunities, which was consistent with the results of this study.

Meanwhile, baby boomers in rural areas showed lower life satisfaction than those in urban areas, which may be due to the different levels of life satisfaction depending on the environmental conditions of urban and rural areas. The residential environment is an important condition that affects life satisfaction [32,33,34]. However, rural areas are currently perceived as “an inconvenient place to live” and “a place that lacks welfare and cultural facilities”, as opposed to a nice place where people would want to live. This leads to a lack of cultural and welfare facilities, such as welfare, healthcare, markets, and recreational facilities, thereby further reducing the quality of life among residents in rural areas [35].

Moreover, residents of rural areas had a lower subjective quality of life or education levels than their urban counterparts, and thus overall life satisfaction was remarkably lower in rural areas than in urban areas [36]. Various environmental conditions that influence life satisfaction were inferior in rural areas compared with urban areas, which served as the main cause of lower life satisfaction. Several other studies [37,38,39] have also shown that residents of rural areas have lower life satisfaction than urban areas.

Second, the RLD of baby boomers had a negative effect on life satisfaction. That is, higher RLD leads to lower life satisfaction. Townsend [1] presented deprivation as a multidimensional concept, dividing it into material deprivation from a lack of nutrition, clothing, and labor, and social deprivation from a lack of social participation and leisure education. Since it was introduced by Stouffer, Suchman, DeVinney, Star, and Williams [40], relative deprivation has evolved as a theory in social science [41], and Hwang and Kim [5] studied the conceptualization and scale development of RLD with a focus on the lack of overall leisure resources among the elements of multidimensional deprivation. However, studies on RLD are still in the early stages in Korea. Accordingly, this study approaches the comprehensive concept of relative deprivation by focusing on the lack of overall leisure resources among the elements of multidimensional deprivation.

Previous studies on relative deprivation and life satisfaction have reported significant findings. Seo [42] studied 936 social media users and discovered that relative deprivation had a negative effect on life satisfaction. Moreover, participants’ negative emotions indirectly reduced quality of life through the path of perceiving their financial situation in Korean society much more negatively than it was in reality. Leisure activities on social media such as Instagram, blogs, and Facebook are increasing because it has become easy to access the Internet regardless of location. Some side effects of this are bluffing and self-displays on social media that exacerbate relative deprivation, thereby reducing leisure activities and life satisfaction [43]. Currently, this has some significant implications, as an increasing number of leisure activities now take place in contactless ways because of the COVID-19 pandemic.

Kim [44] discovered that social, health and medical, and occupational and financial deprivation had a negative effect on life satisfaction and emphasized the need to provide policies and services that reflect the types of deprivation and characteristics of individuals. It is also necessary to set a baseline to determine the kinds of multidimensional deprivation that one can experience in everyday life, apart from the concept of poverty defined in terms of income. Lee [45] also pointed out that relative deprivation is a factor that decreases life satisfaction, and Yang [46] claimed that relative deprivation has the biggest effect on life satisfaction, emphasizing once again that relative deprivation is a highly negative factor that reduces life satisfaction.

People experience relative deprivation when they compare themselves with similar others and feel that something is lacking in their own lives [41] A lack of leisure resources, which are universal social services provided by the central or local governments, may cause dissatisfaction with life and reduced quality of life by keeping citizens from fulfilling their basic needs and rights to participate in leisure activities [47,48,49] Leisure resources are human and material resources that enable leisure activities, including individual leisure resources such as physical health, economic resources, and leisure time, personal leisure resources such as friends and family, and tangible and intangible social leisure resources such as leisure information and facilities such as welfare institutes and parks [50].

Baby boomers in Korea are people who have worked especially hard as the pioneers of economic growth, who still spend most of their time working and concentrating on work [51]; thus, they may have perceived a higher level of deficiency in leisure time compared to other generations. Baby boomers also have relatively higher expectations for happiness because they have been through difficult times that require responsibility and duty due to the aftereffect of economic recessions in the past [52], thereby showing higher needs for overall leisure resources.

Unfortunately, social interest in leisure in today’s Korean society is growing, which may increase relative deprivation in leisure life as people compare themselves to others [31]. Thus, to promote life satisfaction and happiness of baby boomers in this environment, it is necessary to find ways to minimize RLD by expanding diverse and high-quality leisure resources.

Third, baby boomers’ place of residence moderated the relationship between RLD and life satisfaction. More specifically, higher RLD led to lower life satisfaction of baby boomers living in rural areas, whereas RLD did not affect the life satisfaction of baby boomers in urban areas. Therefore, place of residence was shown to be an important moderating variable in the relationship between RLD and life satisfaction.

In Korean society, urban and rural areas are historically, socially, and culturally homogeneous, but there are significant differences in terms of physical facilities, living environment, education, and healthcare depending on the place of residence [29]. Urban areas are better equipped with these infrastructures and facilities, whereas rural areas lack various environmental conditions conducive to life satisfaction compared to urban areas.

Baby boomers living in urban and rural areas share similarities because they have lived throughout the same period, but while baby boomers in urban areas experience various leisure lifestyles, those in rural areas have dull, monotonous leisure lifestyles [17]. Furthermore, people in rural areas lack access to proper healthcare compared to those in urban areas, thereby being alienated from healthcare services, and also live in poor environmental conditions in terms of leisure welfare services that maintain and promote physical, mental, and social health and improve the quality of life [17].

In other words, as mentioned previously, rural areas lack leisure resources such as various kinds of infrastructure and general facilities related to leisure activities compared to urban areas. Moreover, residents of rural areas have lower incomes and spend more time working compared to those in urban areas, thereby experiencing insufficient leisure time. Kim and Kim [52] claimed that baby boomers’ lack of leisure time and the burden of leisure expenditure serve as constraints on participating in various leisure activities. Kwak and Hong [30] pointed out that insufficient leisure time and income are the causes of leisure inequality. Furthermore, residents of rural areas face much more inconvenience in accessing and participating in leisure activities, which may intensify relative deprivation even more [31].

This study confirmed that place of residence is a key variable that moderates RLD and life satisfaction. In particular, higher RLD in rural areas with a lack of overall leisure resources led to lower life satisfaction. Based on the above findings, this is because of the regional gap in leisure resources such as environmental conditions, leisure and welfare services, and accessibility to leisure facilities depending on the place of residence, which is insufficient in rural compared to urban areas. This conditional difference between urban and rural areas may serve as a critical factor that reduces or increases RLD, and awareness of life satisfaction may also vary depending on RLD [44]. Therefore, guaranteeing equal access to resources and opportunities by improving the physical and socioeconomic environment of deprived areas, will thereby minimize the RLD of citizens and improve their life satisfaction. However, it is necessary for individuals to achieve a work-life balance by developing the ability to efficiently use their different amounts of leisure time and resources available.

In summary, this study is meaningful in that it designed and investigated the effect of relative leisure deprivation on quality of life in Korea by comparing it between regional disparity at a time when it remains in a very early stage in Korea. In addition, the number of samples of 412 and the high validity and reliability of the survey tool were secured even though they were specific targets of the baby boomer generation. However, the fact that the sub-factors of relative leisure deprivation were not analyzed in detail and that individual psychological aspects were not considered due to the limitations of quantitative research suggests the direction of subsequent studies.

Therefore, based on this study, further research requires various analyses and specific interpretations depending on the characteristics of each RLD subfactor. In addition, a more in-depth approach must be taken to overcome the limitations of quantitative research, such as in-depth interviews and observations.

Furthermore, to narrow the regional gap, the central or local governments must seek a balanced distribution of leisure resources to remote areas and underdeveloped environments so that residents even at the “ri” level (the smallest administrative unit) can enjoy their benefits, instead of concentrating resources on areas with a high floating population. Moreover, it is necessary to effectively expand and supply leisure resources and switch policies to leisure life and welfare, in which everyone can easily and conveniently enjoy leisure in everyday life beyond leisure welfare.

## 5. Conclusions

The following conclusions were drawn. First, baby boomers living in rural areas showed higher RLD and lower life satisfaction than those living in urban areas. Second, the RLD of baby boomers had a negative effect on life satisfaction. Third, higher RLD of baby boomers living in rural areas led to lower life satisfaction, whereas RLD of baby boomers living in urban areas did not affect life satisfaction.

According to the Framework Act on the Promotion of Leisure of Citizens in Korea, the central and local governments are supposed to develop and disseminate leisure programs, provide leisure information, offer leisure education, expand leisure facilities and venues, and guarantee leisure for even the disadvantaged without alienation. However, the results of this study show otherwise.

## Figures and Tables

**Figure 1 ijerph-18-12905-f001:**
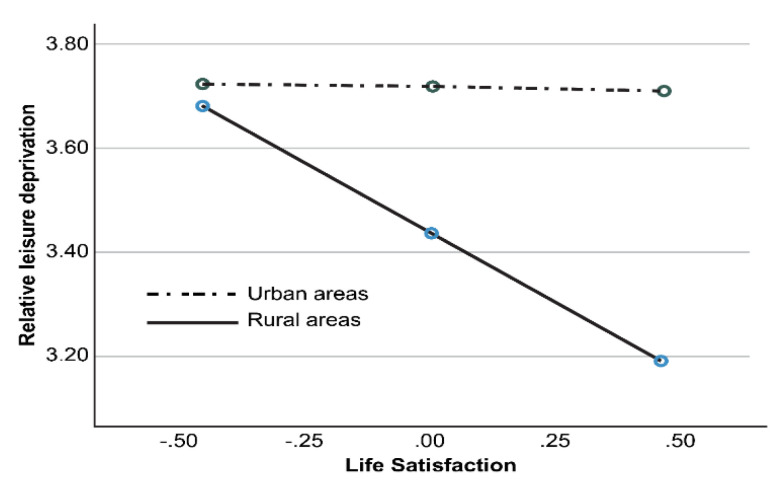
Moderating effect of place of residence on the relationship between relative leisure deprivation (RLD) and life satisfaction.

**Table 1 ijerph-18-12905-t001:** Demographic characteristics.

Variable	Category	*n*	%	Variable	Category	*n*	%
Gender	Male	183	44.4	Place of residence	Rural areas	197	47.8
					Urban areas	215	52.2
	Female	229	55.6				
Year born (M ± SD)		1958.78 ± 2.72		Monthly household income (won)	Below 2 million	61	14.3
Marital status	Single	19	4.6				
					2−2.99 million	62	15.0
	Married	347	84.2				
					3−3.99 million	128	31.1
	Bereaved/ divorced	46	11.2				
					4−4.99 million	107	26.0
					5 million or above	56	13.6
Occupational status	Employed	293	71.1	Educational background	Middle school graduate or lower	97	23.5
					High school graduate	198	48.1
	Unemployed	119	28.9				
					Junior college or higher	117	28.4
Total		412	100.0	Total		412	100.0

**Table 2 ijerph-18-12905-t002:** Survey composition.

Category (Number of Questions)		Example of Questions
Socio-demographic characteristics		Gender, age, marital status, job status, residential area, monthly income, educational background
Relative Leisure Deprivation Scale (18)	Egoistical (3)	I lack leisure costs compared to others. I don’t have much leisure time compared to other people. I am not provided with leisure information compared to others.
	Resourceful (3)	The place where I live lacks leisure programs compared to other regions. Where I live, there is not enough leisure space compared to other areas. The place where I live lacks leisure facilities compared to other areas.
	Cognitive (6)	Compared to other regions, where I live, there is a lack of various leisure spaces, so I am at a disadvantage. Compared to other people, I am at a disadvantage because I am not provided with various leisure information. Compared to other regions, where I live, I am at a disadvantage because of the lack of various leisure programs. Compared to other regions, where I live, I am at a disadvantage because of the lack of various leisure facilities. Compared to other regions, where I live, there is a lack of leaders for various leisure activities, so they are at a disadvantage. I am dissatisfied with the lack of leisure costs compared to others.
	Emotional (6)	I am angry because I am being penalized for not being provided with leisure information compared to others. I am angry because where I live, I am at a disadvantage because of the lack of leisure facilities compared to other areas. I am angry because where I live, I am being penalized because of the lack of leisure programs compared to other regions. I am angry because the place where I live is being penalized for lack of leisure space compared to other areas. I get angry because I don’t have enough leisure time compared to other people. I get angry because I don’t have enough leisure costs compared to other people.
Life Satisfaction (6)		I am very satisfied with my life. The conditions of my life are very good. All in all, my life is close to my ideal. So far, I have achieved the important things I want in life. If I were to be reborn, I would keep my life pretty much the same.

**Table 3 ijerph-18-12905-t003:** Fit of the measurement tool.

Variables	*χ*^2^/*df*	GFI	CFI	TLI	RMSEA
Standard	≤0.30	≥0.90	≥0.90	≥0.90	≤0.10
First model	691.01/183	0.881	0.890	0.871	0.074
Modified model	422.34/124	0.923	0.912	0.904	0.069

**Table 4 ijerph-18-12905-t004:** Difference in relative leisure deprivation and life satisfaction depending on place of residence.

Dependent Variable	Independent Variable	*n*	M ± SD	*t*
Relative leisure deprivation	Rural areas	197	3.21 ± 0.41	5.982 ***
	Urban areas	215	2.95 ± 0.46	
Life satisfaction	Rural areas	197	3.36 ± 0.71	−6.106 ***
	Urban areas	215	3.72 ± 0.46	

*** *p* < 0.001.

**Table 5 ijerph-18-12905-t005:** Moderating effect of place of residence.

Path	*β*	*S.E.*	*t*	*p*
Constant	3.4336	0.0431	79.6561	0.000
Relative leisure deprivation (RLD)	−0.5391	0.1006	−5.3576	0.000
Place of residence	0.2832	0.0592	4.7844	0.000
RLD x place of residence	0.5240	0.1317	3.9784	0.000
*R*^2^ increase due to interaction	*R*^2^-change		*F*	*p*
	0.144		22.8148	0.000

**Table 6 ijerph-18-12905-t006:** Conditional effects due to the relationship between relative leisure deprivation and life satisfaction.

Place of Residence	Effect	*SE*	*t*	LLCI ^1^	ULCI ^2^
Rural areas (0)	−0.5391	0.1006	−5.3576 ***	−0.7369	−0.3413
Urban areas (1)	−0.0151	0.0850	−0.1778	−0.1822	0.1519

^1^ Lower limit confidence interval of 95%. ^2^ Upper limit confidence interval of 95%. Dependent variable: life satisfaction. *** *p* < 0.001.

## Data Availability

The data presented in this study are available on request from the corresponding author.

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
