# Peer review of "Effects of Relative Leisure Deprivation between Regions on Life Satisfaction in Koreans: Focusing on Baby Boomers"

_ijerph, 2021, doi:10.3390/ijerph182412905_

Round 1

Reviewer 1 Report

Thank you for the chance to review this manuscript investigating the importance of leisure welfare to the life satisfaction among baby boomers generation. This study is a timely and interesting work which demonstrated the consequences of inadequacy of leisure and raised the awareness of the significance of leisure welfare to public. However, several issues are needed to be addressed.

  1. Baby boomer generation was selected as the investigated subjects. The reasons of why selecting this group of people is not persuasive enough. It is believed that X, Y, Z generations in different regions may also suffer from RLD which may impact their life satisfaction. Some strong evident has to be provided to show that baby boomer generation is the group with a strong association with RLD.
  2. The process of how you selected the 412 participants is missing.
  3. What sampling method did you used for selecting the 412 participants? How to ensure the sample representation?
  4. A table with all the interview questions had better to be demonstrated.

Author Response

The revision was completed by reflecting all the review contents. Thank you for considering even the smallest details so that it can be an in-depth study. In future research, we will strive to become more advanced research. thank you.

Reviewer 2 Report

Thank you for the opportunity to review this interesting manuscript. I have a few minor comments for the authors to consider:

  • Abstract:
    • Lines 13-19: Please include the actual results, e.g. mean differences of RLD (and life satisfaction score), 95% CI, etc
  • Introduction:
    • Line 83: Please clarify if the ‘relative regional disparities’ refer to the economical or developmental disparities. It would be good to relate the disparities to leisure infrastructure, life satisfaction (e.g. in line 86 onwards).
    • Please justify the reason of studying the baby boomers by relating to the health impact, and public health implications.
  • Methods:
    • Line 126: Please add a sentence to clarify the recruitment was convenience sampling or stratified by regions, etc. The authors then need to discuss such recruitment method on the generalisability of the findings.
    • Lines 128-129: Please provide details on how the measurement items of the preliminary survey were verified. For instance, Was it through a panel of expert on the face validity and content validity? Did the authors assess the test-retest reliability and/or internal consistency of the items?
    • Line 130: The authors stated that the survey was “commissioned to Rende Research”. Please clarify the recruitment method (e.g. convenience sampling or gender-stratified through a certain database), and state the inclusion/exclusion criteria applied to contact the 412 participants.
    • Line 136: Please revise the sentence “A questionnaire was used as the measurement tool...” to “Following validation of the preliminary survey (see Section 2.1”), a questionnaire was used as the measurement tool….”
    • Line 174: Please clarify that the t-test was independent samples t-test, and outline how the assumptions (e.g. normality of the outcome variables: RLD and life satisfaction) of independent samples t-test were assessed.
    • Line 175: Please clarify that the place of residence included 2 categories. The authors could revise the phrase to “….depending on the place of residence (rural/urban areas).”
    • Line 178: Please add that 5,000 bootstrap replications were applied.
  • Results:
    • Lines 191-199 (including Table 4): These results can be deleted because it is not primarily related to the hypotheses. Instead, the authors could rephrase lines 176-177 to “Pearson’s correlation analysis was conducted to determine the correlation between the variables, and there was no multicollinearity because the correlation coefficient was no higher than 0.8 (and cite a reference to support your rationale in using the cut-off of 0.8 for multicollinearity).”
    • After deleting Table 4, please revise the numbering of Tables 5 and 6 to Tables 4 and 5 throughout the manuscript.
  • Discussions:
    • Please include a paragraph on the ‘strengths and limitations’ of the present study. For instance, the authors need to address the strengths/limitations of their study design (cross-sectional study, which can affect the investigation around temporal effect), recruitment method (random or non-random sampling technique, which can affect the representativeness of the findings), validity and reliability of the measurement tool (which can affect measurement bias), etc.
    • The authors may move the suggestions (lines 377-388) listed in the Conclusion (Section 5) to the end of the Discussion section (last paragraph).
  • Conclusion:
    • Lines 361-366: These 3 sentences are not concluding remarks. Please delete them.
    • Lines 389-391: The sentence about ‘give meaning’ is vague. Please delete this sentence.

Author Response

(The authors gave the same response as above.)

Reviewer 3 Report

Abstract
More information on the questionnaire should be specified (number of questions, etc).
Some limitation of the study should be presented.
Introduction
Adequate

Methods
Adequate

Results
It is advisable to present the results in blocks of content.

Discussion 
Adequate

Conclusion and Suggestions
Some limitations of the study should be presented.

References
It is advisable to include more current references (last 5 years).

Author Response

(The authors gave the same response as above.)
